# Effect of Leukocyte- and Platelet-Rich Fibrin on Peri-Implant Mucosal Thickness in Edentulous Patients Treated with Mandibular Implant-Retained Overdentures: A Randomized Controlled Trial

**DOI:** 10.3390/jcm14196917

**Published:** 2025-09-29

**Authors:** Ximena Moreno, Patricio Neira, Franz J. Strauss, María Ignacia Mery, Reinhard Gruber, Franco Cavalla

**Affiliations:** 1San Camilo Hospital, San Felipe 8940000, Chile; xmorenor@odontologia.uchile.cl (X.M.); pneira@odontologia.uchile.cl (P.N.); 2Faculty of Dentistry, Universidad de Chile, Santiago 8320000, Chile; mmery@odontologia.uchile.cl; 3Clinic of Reconstructive Dentistry, Center of Dental Medicine, University of Zurich, 8006 Zurich, Switzerland; franz.strauss@zzm.uzh.ch; 4Health Science Faculty, Universidad Autonoma de Chile, Santiago 8940000, Chile; 5Department of Periodontology, Medical University of Vienna, 1090 Vienna, Austria; reinhard.gruber@meduniwien.ac.at; 6School of Dentistry, Universidad Andres Bello, Santiago 7591538, Chile

**Keywords:** dental implants, guided tissue regeneration, jaw, edentulous, platelet-rich fibrin

## Abstract

**Background/Objectives**: The maintenance of peri-implant soft tissue health is critical for the long-term success of implant therapy, particularly in edentulous patients rehabilitated with mandibular overdentures. Leukocyte- and platelet-rich fibrin (L-PRF) has been proposed as an autologous biomaterial to enhance peri-implant tissue quality. This randomized controlled clinical trial evaluated the effect of L-PRF on peri-implant mucosal thickness in edentulous patients treated with mandibular implant-retained overdentures. **Methods**: Edentulous patients received two interforaminal implants to retain a mandibular overdenture and were randomly assigned to a test group (L-PRF applied during surgery) or a control group (standard protocol without L-PRF). Clinical measurements of keratinized mucosal thickness and width were recorded at baseline, 12 weeks, and 24 weeks. Volumetric analyses of soft and hard tissue changes were performed using digital superimposition of STL models. The trial was conducted in accordance with the Declaration of Helsinki and approved by the Scientific Ethics Committee of the Aconcagua Health Service. All participants provided written informed consent. **Results**: A significant increase in keratinized mucosal thickness was observed in the L-PRF group at 12 and 24 weeks compared with baseline (*p* < 0.01). No significant differences were detected between the groups in soft tissue volume (*p* = 0.12) or bone volume (*p* = 0.45). Mucosal width remained stable in both groups throughout follow-up. **Conclusions**: The application of L-PRF at implant placement resulted in a significant gain in peri-implant mucosal thickness, suggesting a soft tissue modulating effect. Enhancing keratinized mucosal thickness during implant surgery may improve peri-implant tissue quality and support long-term stability of mandibular overdentures.

## 1. Introduction

Tooth loss leads to alveolar bone resorption and disruption of the alveolar process architecture. In the anterior mandible, this resorption pattern typically involves both vertical and horizontal bone loss, while in the posterior region, bone loss is predominantly vertical [1,2]. Atrophy in the anterior mandible often results in knife-edge ridges characterized by preserved height but severely reduced width [3], creating suboptimal conditions for oral rehabilitation.

Although conventional removable prostheses remain a reliable and effective solution for edentulous patients, unfavorable residual bone conditions often lead to poor stability, reduced masticatory function, and patient dissatisfaction [4]. Approximately two-thirds of patients wearing conventional mandibular prostheses report problems with retention and stability [5]. A well-established alternative is the implant-retained overdenture, which consists of a removable acrylic complete denture retained by two interforaminal implants. This approach provides improved retention, leading to enhanced stability, function, aesthetics, and quality of life in patients with atrophic ridges [4]. Studies have shown significantly greater masticatory efficiency and bite force in patients using mandibular implant-retained overdentures compared with conventional prostheses. Consequently, these patients also demonstrate better nutritional status, with notable improvements in biochemical and anthropometric nutritional parameters [6].

Advanced age has not been shown to compromise implant survival rates [7]; however, individuals over the age of 65 are more susceptible to developing peri-implant mucositis and peri-implantitis [8]. Therefore, while geriatric patients are generally suitable candidates for implant therapy, it is essential to identify and manage local factors that may increase the risk of peri-implant inflammation [8,9].

Leukocyte- and platelet-rich fibrin (L-PRF) is an autologous biomaterial prepared by centrifugation of the patient’s own blood [10]. Its use is based on the premise that supraphysiological concentrations of growth factors may enhance the early stages of wound healing and tissue regeneration beyond what is achieved under physiological conditions [11]. L-PRF has been associated with improved implant primary stability and enhanced early healing in some clinical models [10,12].

The application of L-PRF has also been proposed to promote the development of a keratinized mucosal band around the implant neck, potentially contributing to improved peri-implant soft tissue quality and long-term implant success [13].

Both the width and thickness of the keratinized mucosa are important parameters in maintaining peri-implant tissue health. A minimum band of keratinized mucosa facilitates oral hygiene, reduces mucosal inflammation, and maintains soft tissue integrity around implants, particularly in removable overdenture protocols [14]. Inadequate keratinized width may result in mucosal mobility, discomfort during function, and increased plaque accumulation—factors that can compromise long-term implant success [14,15]. Similarly, greater mucosal thickness has been associated with enhanced resistance to mechanical and microbial challenges and may help preserve marginal bone levels and reduce the risk of mucosal recession [14]. These considerations are especially relevant in edentulous patients rehabilitated with mandibular overdentures, where mucosal health is critical for prosthesis retention, hygiene maintenance, and patient comfort [16].

In this context, leukocyte- and platelet-rich fibrin (L-PRF) has been proposed as a soft tissue augmentation option. Clinical evidence indicates that its use can increase both the width of keratinized mucosa and the soft tissue thickness, while reducing operative time and postoperative morbidity compared with connective tissue grafts [17]. In alveolar ridge preservation procedures, L-PRF has been associated with greater gains in soft tissue thickness after 6 months [16]. Additionally, systematic reviews suggest that PRF-based interventions may offer clinical outcomes comparable to autogenous grafts in terms of keratinized mucosa width and tissue thickness [14].

The objective of this split-mouth randomized clinical trial was to evaluate the effect of L-PRF on implant stability, peri-implant clinical parameters, keratinized mucosal thickness, and patient-reported post-surgical pain in edentulous elderly patients receiving implant-retained mandibular overdentures.

Null hypothesis: The adjunctive use of L-PRF during implant placement in edentulous elderly patients receiving implant-retained overdentures does not significantly affect implant stability, peri-implant clinical parameters, keratinized mucosa thickness, or patient-reported outcomes measures compared with the control treatment.

## 2. Materials and Methods

### 2.1. Study Design

This split-mouth randomized clinical trial was conducted at the Periodontics Clinic of San Camilo Hospital, San Felipe, Chile. A total of 40 edentulous patients over the age of 60 were screened. Eligible participants had a history of failed treatment with conventional removable mandibular prostheses. Exclusion criteria included systemic or psychological conditions contraindicating implant placement, mandibular extractions within the previous 6 months, heavy smoking, oral anticoagulant therapy, and hemodialysis.

### 2.2. Screening and Consenting

All participants and outcome assessors were blinded to group allocation. The clinical examiner was a calibrated periodontist (X.M.). Randomization was performed using a computer-generated sequence to assign the experimental (L-PRF) side, and sealed envelopes were used to conceal allocation. Ethical approval was obtained from the Scientific Ethics Committee of the Aconcagua Health Service (CEC09/2020), and all participants signed informed consent prior to enrollment. The study was conducted in accordance with the Declaration of Helsinki and was registered at ClinicalTrials.gov (ID: NCT04429373; 9 June 2020).

### 2.3. Outcome Measures

Measured outcomes included implant stability, assessed by Implant Stability Quotient (ISQ) via resonance frequency analysis (Osstell, W&H, Vienna, Austria); keratinized tissue width (KTW), measured as the vertical distance from the mucosal margin to the mucogingival junction; and probing depth (PD). Mucosal thickness (MT) was measured at the mid-buccal point of each implant site using a standardized transgingival probing technique. A sterile #15 endodontic K-file equipped with a silicone stopper was employed as the measuring instrument. After applying a topical anesthetic gel to minimize discomfort, the file was inserted perpendicularly through the mucosa until firm contact with the underlying bone was achieved. The silicone stopper was then positioned flush with the external mucosal surface, and the distance from the file tip to the stopper was measured with a digital caliper to the nearest 0.1 mm. Each site was measured twice, and the mean value was recorded. To ensure reproducibility, all measurements were performed under standardized conditions of lighting and 2.5× magnification. Examiner calibration was conducted prior to the study on five non-study patients, resulting in an intraclass correlation coefficient (ICC) > 0.90. This approach provides a direct, minimally invasive, and reproducible method for quantifying mucosal thickness.

Cone-beam computed tomography (CBCT) scans (PLANMECA ProMax, Helsinki, Finland) were obtained preoperatively and 6 months post implant installation. Images were analyzed using Blue Sky Plan 4 (BlueSky Bio, Grayslake, IL, USA) by a researcher blinded to the treatment allocation. Three-dimensional models were generated via semi-automatic segmentation and aligned using six anatomical landmarks, following a validated protocol [18,19]. STL surface models were exported and analyzed using MeshMixer 3.5.474 (Autodesk, San Francisco, CA, USA). Additionally, conventional silicon impressions were obtained and later scanned using a Medit T710 scanner (Medit, Seoul, Republic of Korea) to acquire STL models at baseline and at 6 months post implant placement.

Patient-reported postoperative pain was evaluated using a visual analog scale (VAS, 0–10), where 0 indicated “no pain” and 10 indicated “worst pain imaginable.” Each participant received standardized written and verbal instructions on how to complete the VAS. Assessments were performed at 7 and 14 days post-surgery during follow-up appointments by a blinded examiner. Patients marked their pain intensity on a 10 cm horizontal line, and the score was measured in millimeters from the left end of the scale to the patient’s mark. These values were recorded and included in the statistical analysis as continuous variables.

### 2.4. Surgical Protocol

L-PRF was prepared following established protocols [11,20]. Briefly, two 10 mL sterile plastic tubes made of polyethylene terephthalate (PET) with siliconized inner walls (16 mm × 100 mm; red cap, BD Vacutainer) were filled with peripheral blood collected from the median cubital vein and centrifuged at 400× *g* (≈2700 rpm, IntraSpin™, Boca Raton, FL, USA, centrifuge, Intra-Lock) for 12 min at room temperature (Intra-Lock, Vista, CA, USA). The resulting fibrin clots were carefully separated from red blood cell fractions by gently scraping off the red portion and compressed using an L-PRF box (Xpression, IntraLock, Vista, CA, USA) for 5 min to obtain membranes. All surgical procedures were performed by experienced periodontists (P.N., F.C.). A supracrestal full-thickness flap with a periosteal releasing incision was elevated to expose the underlying bone. Then, an osteotomy to place two interforaminal implants following the manufacturer’s instructions was performed (Tapered Internal, BioHorizons, Birmingham, AL, USA). The implants were installed at the paramedian mandibular edentulous ridge, at both sides of the central line, spaced ≤ 15 mm apart. ISQ values were measured immediately after implant placement. Two L-PRF membranes were placed near the implant platform over the buccal bone at the experimental site before wound closure using simple sutures (Monosyn 5/0, B. Braun, Melsungen, Germany). A healing cap was installed on every implant, and follow-up ISQ measurements were taken at 8, 12, and 24 weeks.

### 2.5. Statistical Analysis

Statistical analysis was performed by a blinded independent researcher. Quantitative variables were expressed as means and standard deviations. Data distribution was assessed by visual inspection and the Shapiro–Wilk test. Paired comparisons were conducted using Student’s *t*-test or the Wilcoxon signed-rank test for non-parametric data. Statistical significance was set at *p* < 0.05. Analyses were performed using Stata v14 (StataCorp, College Station, TX, USA).

## 3. Results

A total of 48 patients were screened for eligibility. Eight were excluded; three declined to participate, and five did not meet the inclusion criteria. Thus, 40 patients (26 women; mean age 68.4 ± 4.95 years) were enrolled, and 36 completed the 6-month follow-up (Figure 1). In total, 80 implants were placed (2 per patient): 50 with a diameter of 3.4 mm and 30 with a diameter of 3.8 mm. Implant lengths were 10.5 mm (*n* = 50) and 12 mm (*n* = 30), selected according to the anatomical characteristics of the recipient sites.

Of the participants, 18 (45%) were classified as ASA I and 22 (55%) as ASA II. Thirty-one patients (77.5%) were non-smokers, while nine reported smoking fewer than ten cigarettes per day. Regarding prosthesis history, 33 participants (83%) previously used mandibular removable dentures; however, 63% reported dissatisfaction, mainly due to functional limitations. Masticatory performance, assessed with the Leake Index, was rated as low or moderate in all patients.

Most implants (88%) achieved optimal primary stability (insertion torque > 35 Ncm; ISQ > 60). Initial ISQ values averaged 73.1 ± 5.8, with comparable baseline values between the experimental and control sides (72.4 ± 9.1 vs. 73.4 ± 7.3). At week 8 (T1), mean ISQ values were 75.1 ± 7.0 (experimental) and 76.3 ± 6.5 (control). At week 12 (T2), a significant difference was observed (83.5 ± 9.4 vs. 86.7 ± 7.8; *p* = 0.04). By week 24 (T3), ISQ values were 84.0 ± 8.1 and 84.6 ± 7.7, with no significant differences (Figure 2).

Sixteen postoperative complications were recorded, mainly hematomas (*n* = 15). One patient developed gingival overgrowth at 8 weeks (control side), which required surgical excision. Overall, complication rates did not differ significantly between the groups (*p* = 0.08).

PD was measured at T2 and T3 and KTW at T0, T2, and T3. No significant between-group differences were detected at any time point. At T2, PD was slightly lower in the experimental group (1.61 ± 0.5 mm) than in controls (1.76 ± 0.6 mm; *p* = 0.6). The difference was not significant and was not maintained at T3 (Table 1).

Baseline MT values were comparable between groups. A significant increase in MT was observed in the experimental group at both T2 and T3, indicating a favorable effect of L-PRF on soft tissue thickness (Figure 3). The mean difference between control and L-PRF was 0.26 mm at T2 (*p* = 0.012) and 0.71 mm at T3 (*p* < 0.001).

Volumetric analysis using CBCT reconstructions was performed in the 36 patients who completed follow-up. No significant differences were observed in bone volume between baseline (6533.35 ± 1825.8 mm^3^) and T3 (6425.90 ± 1603.25 mm^3^; *p* = 0.45). Similarly, soft tissue volume showed minimal variation (366.68 ± 30.03 mm^3^ at T0 vs. 362.89 ± 26.97 mm^3^ at T3; *p* = 0.12).

Postoperative pain assessed with the VAS was generally low. At day 7, mean scores were 1.6 ± 2.3 in the experimental group and 1.2 ± 2.0 in the control group. On day 14, scores decreased to 0.5 ± 1.2 and 0.1 ± 0.4, respectively, with no significant differences between groups (*p* = 0.12).

## 4. Discussion

This study demonstrates that the application of L-PRF significantly increased mucosal thickness around interforaminal implants after 12 and 24 weeks. Consequently, the null hypothesis was partially rejected. This effect was consistent and represents the most clinically relevant findings of our study. An increase in MT may improve the quality of the peri-implant mucosal seal [21,22], thereby promoting peri-implant health [23] and reducing long-term biological complications associated with plaque accumulation, inflammation, and trauma from overdenture movement.

In contrast, no clinically significant differences were found in implant stability between the experimental and control groups at any of the time points, except at 12 weeks, where a modest but statistically significant difference was observed. These results are consistent with previous studies indicating that ISQ values increase progressively over time as a marker of successful osseointegration [24]. Recent studies have demonstrated that PRF applications may enhance early implant stability, with significantly higher ISQ values at 1 week and 1 month post implant placement compared with controls [25]. In a study by Guan et al., implants treated with PRF showed higher stability at 12 weeks (mean ISQ 74.5) than controls (mean ISQ 70.8), which aligns with our findings of transient ISQ differences at 12 weeks [26].

The greater MT observed in the experimental group supports prior evidence on the positive effects of L-PRF in maintaining soft tissue architecture. A recent systematic review of injectable PRF (i-PRF) for non-surgical gingival augmentation concluded that i-PRF can significantly increase gingival thickness and, in some cases, keratinized tissue width, while also providing superior patient-reported comfort and aesthetic outcomes compared with traditional grafting techniques [27]. However, protocols for i-PRF application are not yet standardized. While some RCTs used one, others applied three or four sessions at 7–10-day intervals, complicating direct comparisons of outcomes and highlighting the need for protocol optimization in future studies.

By contrast, a recent study reported decreased mucosal thickness following the use of L-PRF [28,29]. This discrepancy may be related to differences in healing mode, as all patients in that study experienced soft tissue dehiscence and healed by second intention, whereas in our study, primary wound closure was achieved in all cases. Moreover, another clinical study [30] showed that combining L-PRF with subepithelial connective tissue grafts resulted in greater gains in keratinized tissue thickness compared with connective tissue grafting alone, suggesting a potential synergistic effect.

Other clinical parameters such as probing depth (PD) and keratinized tissue width (KTW) did not differ significantly between the groups, in agreement with previously published data [31]. Likewise, volumetric analyses revealed no significant differences in hard or soft tissue volumes between the groups at 6 months. The analysis of soft tissue volume using STL files also showed no significant changes, which may reflect methodological limitations inherent to digital subtraction techniques [32,33,34].

Another important consideration is the lack of standardization across clinical protocols involving PRF. Recent trials have differed substantially in terms of the type of PRF used (L-PRF, A-PRF, i-PRF, or T-PRF), centrifugation settings, and the number of applications. For instance, in non-surgical gingival augmentation protocols, some randomized trials have applied a single session of i-PRF, while others used up to three or four sessions at weekly intervals [27,35]. Despite these variations, the clinical use of L-PRF offers several practical advantages. Its preparation is straightforward, cost-effective, and free of additives such as anticoagulants or chemical manipulation [36]. This facilitates its integration into routine practice without additional burden for clinicians or patients.

From a biological perspective, the positive effect of L-PRF on mucosal thickness may be explained by its gradual release of growth factors such as PDGF, VEGF, and TGF-β1, which enhance angiogenesis, fibroblast proliferation, and collagen deposition [37,38]. The fibrin matrix also serves as a scaffold that supports cellular migration and neovascularization. Previous in vitro and in vivo studies [38] have shown that L-PRF membranes sustain growth factor release for up to 7–14 days, which may explain the enhanced tissue maturation observed clinically. These biological mechanisms provide a plausible rationale for the consistent increase in mucosal thickness observed in the present study [39,40].

Future studies should evaluate the long-term stability of the increased mucosal thickness observed with L-PRF and its impact on peri-implant health and prosthetic outcomes. Research in different clinical scenarios and patient populations, combined with improved digital measurement methods, is needed to confirm these findings. Comparative trials with other platelet concentrates and mechanistic investigations may also help clarify the biological basis and relative effectiveness of L-PRF in enhancing peri-implant soft tissue outcomes.

## 5. Limitations

This study has some limitations that should be acknowledged. First, although the split-mouth design was chosen to minimize inter-individual variability and to increase statistical power with a limited number of participants, this design inherently carries the risk of carry-over effects between experimental and control sites. Local biological responses to L-PRF may influence adjacent tissues, thereby potentially attenuating the true magnitude of the differences between groups.

Second, the study sample consisted exclusively of elderly edentulous patients rehabilitated with mandibular overdentures. While this represents a highly relevant clinical population, the results cannot be directly extrapolated to younger individuals, partially edentulous patients, or cases involving different implant sites and prosthetic designs. Moreover, the relatively small sample size limits the ability to detect subtle differences in some clinical outcomes and increases the risk of type II error.

Third, the follow-up period of 24 weeks, although sufficient to capture short- and medium-term soft tissue changes, does not provide information about the long-term stability of the observed mucosal thickness gain or its influence on peri-implant health and prosthetic survival. Longer observational periods are needed to determine whether the soft tissue benefits of L-PRF translate into improved biological and prosthetic outcomes over time.

Finally, the study did not assess biological markers or histological outcomes that could elucidate the underlying mechanisms of the observed clinical changes. Such data would help clarify the pathways through which L-PRF contributes to mucosal thickening and whether these effects are primarily related to angiogenesis, fibroblast activation, or collagen maturation. Taken together, these limitations indicate that while our findings support the potential of L-PRF to enhance peri-implant mucosal thickness, further research with larger, more diverse populations, longer follow-up, and complementary biological analyses is required to fully establish its clinical effectiveness.

## 6. Conclusions

L-PRF use during implant placement was associated with increased peri-implant mucosal thickness, which may improve soft tissue outcomes in the rehabilitation of edentulous patients with mandibular overdentures. Further studies are needed to confirm these findings and to clarify their long-term clinical implications.

## Figures and Tables

**Figure 1 jcm-14-06917-f001:**
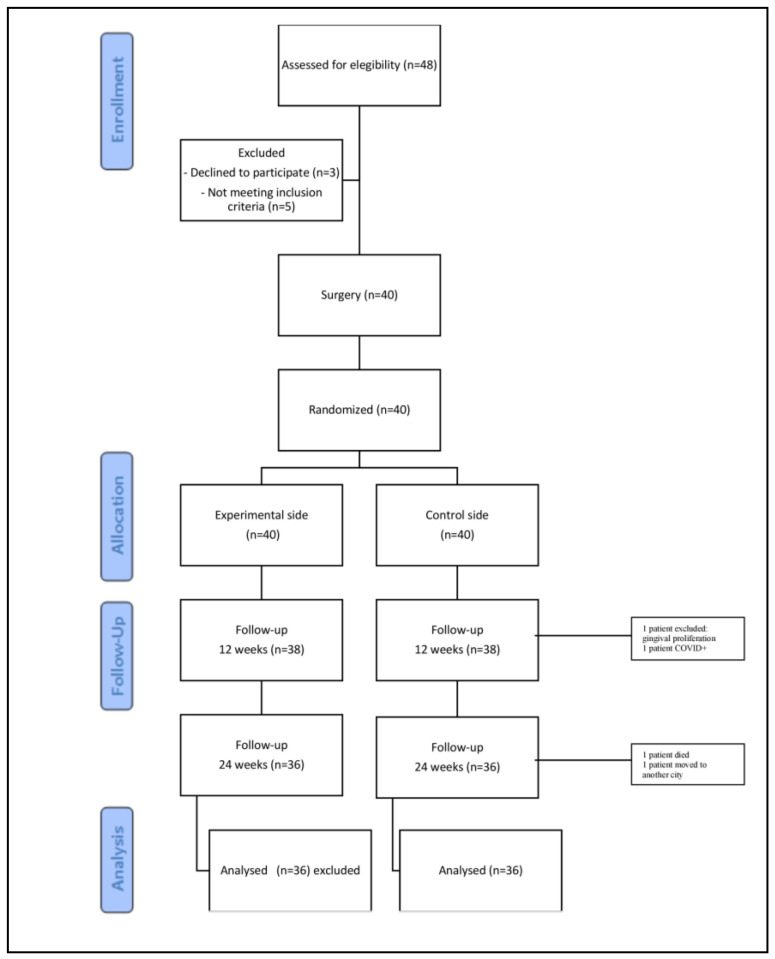
CONSORT flow diagram.

**Figure 2 jcm-14-06917-f002:**
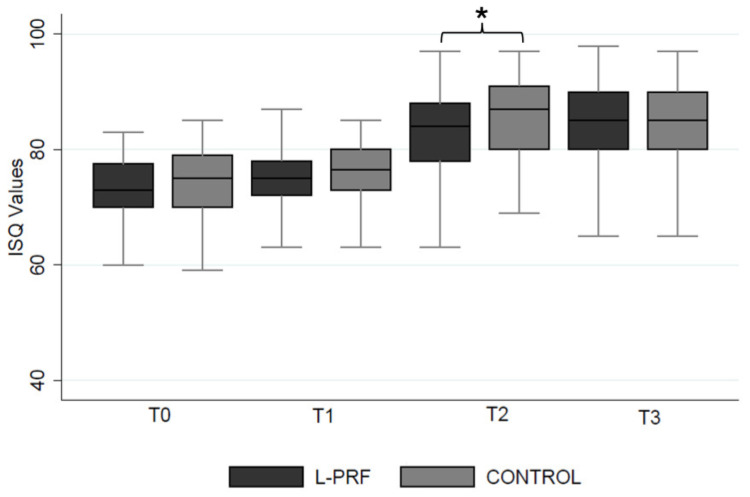
Implant stability at baseline (T0), 8 weeks (T1), 12 weeks (T2), and 24 weeks (T3). * = *p* < 0.05. ISQ: Implant Stability Quotient; L-PRF: leukocyte- and platelet-rich fibrin.

**Figure 3 jcm-14-06917-f003:**
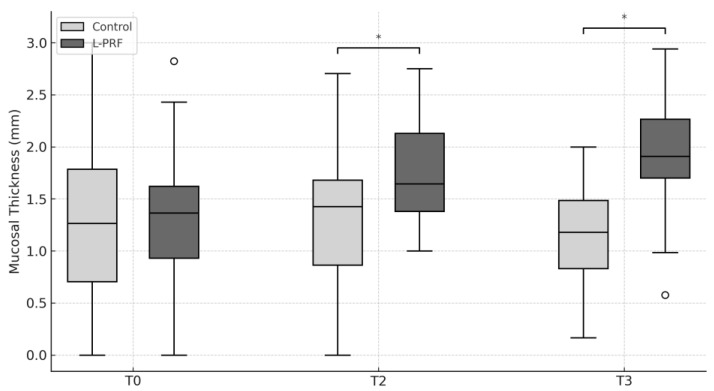
Mucosal thickness (MT) at baseline (T0), 8 weeks (T1), 12 weeks (T2), and 24 weeks (T3). * = *p* < 0.05. L-PRF: leukocyte- and platelet-rich fibrin.

**Table 1 jcm-14-06917-t001:** Clinical parameters of probing depth and keratinized tissue width. T0 = baseline, T2 = 12 weeks post-surgery, T3 = 24 weeks post-surgery.

	L-PRF			Control			*p*-Value		
	**T0**	**T2**	**T3**	**T0**	**T2**	**T3**	**T0**	**T2**	**T3**
PD (mm)		1.6 (±0.5)	1.4 (±0.5)		1.8 (±0.6)	1.4 (±0.5)		0.60	0.46
KTW (mm)	2.7 (±1.6)	2.5 (±1.8)	2.4 (±0.9)	2.5 (±1.8)	2.4 (±0.9)	2.7 (±0.9)	0.53	0.68	0.79

## Data Availability

The data supporting the findings of this study are not publicly available due to patient confidentiality and legal and ethical restrictions. However, the datasets generated and analyzed during the current study are available from the corresponding author upon reasonable request.

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
