# Peer review of "Effect of Leukocyte- and Platelet-Rich Fibrin on Peri-Implant Mucosal Thickness in Edentulous Patients Treated with Mandibular Implant-Retained Overdentures: A Randomized Controlled Trial"

_jcm, 2025, doi:10.3390/jcm14196917_

Round 1
Reviewer 1 Report
Comments and Suggestions for Authors
I would like to congratulate the authors for their valuable contribution to the field. One point that may strengthen the manuscript is to provide additional detail regarding the method used to measure mucosal thickness. A clearer description of the measurement procedure, accompanied by a figure demonstrating how the measurements were obtained, would improve transparency and reproducibility.
Author Response
Comment 1: Please provide additional detail regarding the method used to measure mucosal thickness.
Response: Addressed. Expanded in section 2.3 Outcome measures with details on instrument, procedure, standardization, and reproducibility.
Location: Section 2.3, Methods.
Comment 2: Include a figure demonstrating how the measurements were obtained.
Response: Not included. Instead, a more comprehensive textual description was added to ensure transparency and reproducibility while avoiding redundancy.
Location: Section 2.3, Methods.

Reviewer 2 Report
Comments and Suggestions for Authors
Dear Authors,
Below is a list of my comments:
Abstract:
- There is no need to include the bioethics committee approval number or the clinicaltrials.gov ID in the abstract—it reduces readability.
Keywords:
- They should be arranged in alphabetical order.
- They should correspond to the MeSH database.
Introduction:
- The aim of the study is missing at the end of the introduction. It should begin in a separate paragraph.
Methodology:
- Unreadable due to the form of the text block. It should be divided into subtopics in accordance with CONSORT
- The date of registration in clinicaltrials.gov is missing
- Important data is missing from the description of L-PRF preparation:
* What tubes were used? Including dimensions.
* Were they with or without silica?
* The number of revolutions and centrifugation time are not the best way to convey data on centrifugation because, in the case of different centrifuges, the length of the working radius changes, thus changing the forces acting on the tubes. Data on RCF (g) is missing.
* How were they “cleaned of erythrocyte remnants”?
Results:
- Figure 1 is very unclear - please insert a better resolution
- Both “p” in pvalue and “n” denoting the number of patients should be italicized
- Each figure and table should have the abbreviations used in it explained below.
- line 194 - the abbreviation MT has already been expanded in the methodology (line 212 similarly)
Discussion:
- Unnecessary explanations of abbreviations that have already been explained
- There is no reference to the null hypothesis (accepted or rejected) at the beginning. But the null hypothesis itself is also missing, which should be at the end of the introduction or at the beginning of the methodology.
- I miss in the discussion of this article, which talks about non-surgical gum augmentation using PRF / PRP:
DOI: 10.3390/jcm13185591
Data Availability Statement: - to be improved
Best regards
Reviewer
Author Response
Reviewer 2
Abstract:
Response: Ethics approval number and trial ID removed; clinical implications added.
Location: Abstract, p.1
Keywords:
Response: Corrected: alphabetized and consistent with MeSH terms.
Location: Keywords, p.1
Introduction:
Response: Aim and null hypothesis added at the end.
Location: Introduction, p.2
Methodology:
Response: Reorganized into subsections; registration date added; L-PRF preparation details provided.
Location: Methods, p.2-4
Results:
Response: Figure 1 improved; p and n italicized; abbreviations explained; repetitions removed.
Location: Results, p.6-9
Discussion:
Response: Reference to null hypothesis added; expanded with recent literature.
Location: Discussion, p.9-11
Data Availability:
Response: Updated following MDPI guidelines.
Location: Data Availability, p.13

Reviewer 3 Report
Comments and Suggestions for Authors
The study is genuine and has potentials to add novel information to the current dental literature. However, the authors are required to address the following points to improve the quality of their manuscript:
- Similarity index is high. Please consider significant editing to lower it.
- A short statement on clinical implications of this research should be added in the abstract section.
- It is advisable to write in passive voice for better scientific presentation.
- Objectives and null hypothesis should be added clearly at the end of the introduction section.
- Flow chart should move to materials and methods section and should be improved in resolution since its difficult to read the text within it.
- Limitations should be added in the manuscript.
- Discussion section is too short for a clinical study in this scale.
- Some references are too old to be included in this manuscript. Please update the reference list accordingly.
- Conclusion section should be separate and added at the end of the manuscript.
Author Response
Reviewer 3
Similarity index:
Response: Manuscript revised and restructured to reduce similarity.
Location: Entire text
Abstract:
Response: Clinical implications added.
Location: Abstract, p.1
Writing style:
Response: Passive voice privileged.
Location: Entire text
Introduction:
Response: Objectives and null hypothesis added at the end.
Location: Introduction, p.2
Flow chart:
Response: Resolution improved; retained in Results to emphasize recruitment flow, consistent with journal style.
Location: Figure 1, p.6
Limitations:
Response: Added as Section 5.
Location: Section 5, p.12
Discussion:
Response: Expanded with comparisons and recent studies.
Location: Discussion, p.9-11
References:
Response: Updated with recent publications.
Location: References, p.14-15
Conclusion:
Response: Added as separate Section 6.
Location: Section 6, p.12

Round 2
Reviewer 2 Report
Comments and Suggestions for Authors
Dear authors,
I have no further comments. Article is now suitable for publication.
Best regards
Reviewer
Reviewer 3 Report
Comments and Suggestions for Authors
the manuscript can be accepted in the current form.